# Prevalence of dermal trypanosomes in suspected and confirmed cases of *gambiense* human African trypanosomiasis in Guinea

Alseny M'mah Soumah[1], Mariame Camara[1], Justin Windingoudi Kaboré[1], Ibrahim Sadissou[2], Hamidou Ilboudo[2,3], Christelle Travaillé[4], Oumou Camara[1], Magali Tichit[5], Jacques Kaboré[6], Salimatou Boiro[7], Aline Crouzols[4], Jean Marc Tsagmo Ngoune[4], David Hardy[5], Aïssata Camara[7], Vincent Jamonneau[2], Annette MacLeod[8], Jean-Mathieu Bart[1,2], Mamadou Camara[1], Bruno Bucheton[1,2], Brice Rotureau[4,7] *

1 Programme National de Lutte contre les Maladies Tropicales Négligées, Ministère de la Santé, Conakry, Guinea, 2 INTERTRYP, Université de Montpellier, CIRAD, IRD, Montpellier, France, 3 Institut de Recherche en Sciences de la Santé - Unité de Recherche Clinique de Nanoro, Nanoro, Burkina-Faso, 4 Trypanosome Transmission Group, Trypanosome Cell Biology Unit, INSERM U1201, Department of Parasites and Insect Vectors, Institut Pasteur, Université Paris Cité, Paris, France, 5 Histopathology Core Facility, Institut Pasteur, Université Paris Cité, Paris, France, 6 Unité de recherches sur les bases biologiques de la lutte intégrée, Centre International de Recherche-Développement sur l'Elevage en zone Subhumide, Bobo-Dioulasso, Burkina Faso, 7 Parasitology Unit, Institut Pasteur of Guinea, Conakry, Guinea, 8 Wellcome Centre for Integrative Parasitology, College of Medical, Veterinary, and Life Sciences, Henry Wellcome Building for Comparative Medical Sciences, Glasgow, Scotland, United Kingdom

* rotureau@pasteur.fr

**Data Availability Statement:** All data are in the manuscript and supporting information files.

## Abstract

The skin is an anatomical reservoir for African trypanosomes, yet the prevalence of extravascular parasite carriage in the population at risk of *gambiense* Human African Trypanosomiasis (gHAT) remains unclear. Here, we conducted a prospective observational cohort study in the HAT foci of Forecariah and Boffa, Republic of Guinea. Of the 18,916 subjects serologically screened for gHAT, 96 were enrolled into our study. At enrolment and follow-up visits, participants underwent a dermatological examination and had blood samples and superficial skin snip biopsies taken for examination by molecular and immuno-histological methods. In seropositive individuals, dermatological symptoms were significantly more frequent as compared to seronegative controls. *Trypanosoma brucei* DNA was detected in the blood of 67% of confirmed cases (22/33) and 9% of unconfirmed seropositive individuals (3/32). However, parasites were detected in the extravascular dermis of up to 71% of confirmed cases (25/35) and 41% of unconfirmed seropositive individuals (13/32) by PCR and/or immuno-histochemistry. Six to twelve months after treatment, trypanosome detection in the skin dropped to 17% of confirmed cases (5/30), whereas up to 25% of unconfirmed, hence untreated, seropositive individuals (4/16) were still found positive. Dermal trypanosomes were observed in subjects from both transmission foci, however, the occurrence of pruritus and the PCR positivity rates were significantly higher in unconfirmed seropositive individuals in Forecariah. The lower sensitivity of superficial skin snip biopsies appeared critical for detecting trypanosomes in the basal dermis. These results are discussed in the context of the planned elimination of gHAT.

**Funding:** This work was supported by the Institut Pasteur (MT, ACr, DH and BR) and Institut Pasteur of Guinea (SB, Aca and BR), the Institut de Recherche pour le Développement (VJ, JMB and BB), the French Government Investissement d'Avenir programme - Laboratoire d'Excellence "Integrative Biology of Emerging Infectious Diseases" (ANR-10-LABX-62-IBEID sub-grant to BR, salary to CT and JMTN) and the French National Agency for Scientific Research (project ANR-18-CE15-0012 TrypaDerm to BR, salary to IS and CT). The funders had no role in study design, data collection and analysis, decision to publish, or preparation of the manuscript.

**Competing interests:** The authors have declared that no competing interests exist.

## Author summary

The skin is a reservoir for African trypanosomes. Here, we conducted a prospective study in Forecariah and Boffa, Guinea, to estimate the proportion of skin-dwelling parasites in the population. Of the 18,916 subjects screened for HAT, 96 were enrolled into our study. Participants underwent a dermatological examination and had blood samples and superficial skin biopsies taken for examination by molecular and immuno-histological methods. In individuals seropositive for HAT, dermatological symptoms were significantly more frequent. Trypanosome DNA was detected in the blood of 67% of confirmed cases and 9% of unconfirmed seropositive individuals. However, parasites were detected in the skin of up to 71% of confirmed cases and 41% of unconfirmed seropositive individuals. After treatment, trypanosome detection in the skin dropped to 17% of confirmed cases, whereas up to 25% of unconfirmed, hence untreated, seropositive individuals were still found positive. Dermal trypanosomes were observed in subjects from both regions; however, the occurrence of itching and the PCR positivity were significantly higher in unconfirmed seropositive individuals in Forecariah. The lower sensitivity of superficial skin biopsies appeared critical for detecting trypanosomes. These results are discussed in the context of the planned elimination of HAT.

## Introduction

With only 565 new cases reported in 2020, *gambiense* human African trypanosomiasis (gHAT) or sleeping sickness, a neglected tropical disease caused by the tsetse-borne protist parasite *Trypanosoma brucei gambiense* (*T. b. gambiense)*, has been targeted for elimination (zero transmission) by 2030 by the World Health Organization (WHO) [1]. This objective has been encouraged by the success of active surveillance efforts that relies on a two-step diagnosis. An initial serological screen is followed by microscope observation of blood, lymph and/or cerebrospinal fluid (CSF) to detect extracellular trypanosomes for confirmation (Table 1).

We have recently reported the presence of *T. b. gambiense* parasites in the extravascular compartment of the mammalian host skin, under experimental conditions in animal models [2, 3], and during the natural progression of the disease in human patients [4]. At least in experimental conditions, substantial quantities of trypanosomes persist within the basal region of the extravascular dermis and can be transmitted to the tsetse vector, even in the absence of

**Table 1. Diagnostic process and group definition.**

| Groups | | Diagnostic process | | | | |
|---|---|---|---|---|---|---|
| | | 1- Serological screening | 2- Serological validation | 3- Parasitological confirmation | 4- Staging | |
| | | CATTwb / RDT | CATTp* | mAECT BC / LN aspirate observation | Parasites in CSF | WBC in CSF |
| Seronegative | | - | | | | |
| | | + | < 1/4 | | | |
| Seropositive | | + | ≧ 1/4 | - | | |
| Confirmed | Stage 1 | + | ≧ 1/4 | + | no | 0–5 |
| | Stage 2 | | | | yes | >5 |

CATTwb / CATTp: card agglutination test for trypanosomiasis on whole blood / plasma; RDT: Rapid Diagnostic Test for HAT; mAECT BC / LN aspirate: mini anion-exchange column technique on buffy coat / lymph node aspirate; WBC: white blood cells; CSF: cerebrospinal fluid

*Highest plasma dilution with a positive result.

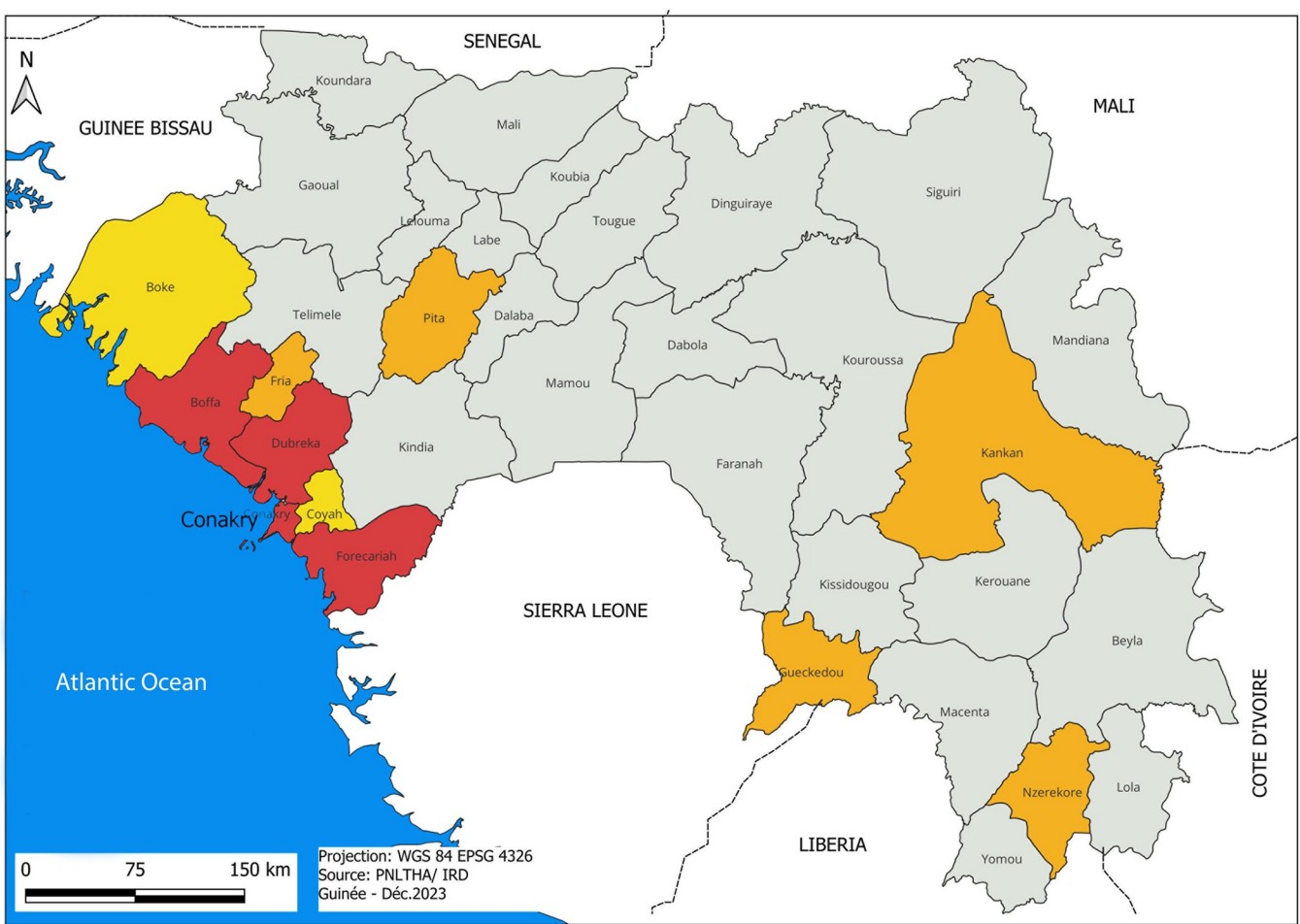

**Fig 1. Map of the HAT foci in Guinea.** This map shows the administrative districts of the Republic of Guinea. Endemic transmission foci in red, old foci with last cases reported before 2004 in orange, zones at risk in yellow. This map was elaborated in-house with QGIS 3.28.12 from an OSM standard layer (www. openstreetmap.org) (adapted from [30]).

detectable parasites in the host's blood [2]. Some seropositive individuals remain unconfirmed for years with no parasite detectable in blood [5]. These untreated individuals can however carry trypanosomes in their extravascular dermis [4]. This raises the question of their potential role of latent reservoirs possibly maintaining the parasite circulation in endemic foci [6].

To refine our understanding of the epidemiological importance of skin-dwelling trypanosomes, we performed a prospective observational study at a larger scale in both the Forecariah and Boffa districts in the Republic of Guinea (Fig 1), which are the most active gHAT foci in Western Africa.

## Methods

### Ethics statement

All investigations were conducted in accordance with the Declaration of Helsinki and fulfil the STROBE criteria. Approval for this study was obtained from the National Ethical Committee of the Republic of Guinea (Study Diag-Cut-THA 032/CNERS/17 and amendment 038/CNERS/19).

## Study enrolment, screening, and case definitions

The overall study design shown in Table 1 was the same as in Camara *et al.* [4]. All subjects enrolled in this study came from villages in the active gHAT foci of the Forecariah and Boffa districts, which are located in the coastal mangrove areas of the Republic of Guinea (Table 2 and Fig 1) [7]. All enrolled subjects were screened from December 2018 to April 2021 in medical surveys performed by the HAT National Control Programme, according to WHO recommendations and as described previously (Table 2) [4, 8]. Individuals were first screened either with the card agglutination test for trypanosomiasis [9] using whole blood samples (CATTwb), or one rapid diagnostic test for HAT on blood (SD Bioline HAT, Abbott Bioline HAT 2.0 or HAT Sero-K-SeT) (Tables 1 and 2). For those individuals who tested positive in the serological screening test, 5ml of blood was collected in heparinized tubes. A two-fold plasma dilution series was used to determine the CATT plasma (CATTp) end titer. In parallel, blood samples of seropositive suspects were centrifuged to obtain the buffy coat layer, which was tested for trypanosomes using the mini-anion exchange centrifugation test (mAECT BC) [10]. A microscopic examination of lymph node aspirate was also performed for suspects with cervical swollen lymph nodes. If trypanosomes were detected using at least one of these parasitological tests, the infected individual (confirmed case) underwent a lumbar puncture and their disease stage was determined by searching for trypanosomes using the modified simple centrifugation technique for CSF and by white blood cell (WBC) counts [11]. *Gambiense* HAT patients were classified as being stage 1 (0–5 WBC/μl and absence of trypanosomes in CSF) or stage 2 (>5 WBC/μl and/or presence of trypanosomes in CSF) and were treated accordingly by the National Control Programme (Table 1). For stage 1 patients, treatment consisted of Pentamidine (intramuscular injection of 4 mg/kg once daily for 7 days in adults) or Fexinidazole (oral doses of 1800 mg fexinidazole once per day on days 1–4 then 1200 mg fexinidazole on days 5–10)[12] or Acoziborole (1 oral dose of 960 mg)[13]. For stage 2 patients, treatment consisted of Nifurtimox-Eflornithine Combination Therapy (NECT) (oral Nifurtimox at 15 mg/kg per day in three doses for 10 days and intravenous Eflornithine (α-difluoromethylornithine or DFMO) at 400 mg/kg per day in two 2h-infusions for 7 days in adults) or Fexinidazole (oral doses of 1800 mg once per day on days 1–4 then 1200 mg on days 5–10)[12] or Acoziborole (1 oral dose of 960 mg)[13]. All parasitologically confirmed cases in this study were diagnosed and treated according to WHO recommendations at that time and to the availability of the different treatments.

In summary, subjects were included in either of these 3 groups according to their diagnostic results (Table 1):

- Seronegative: negative in CATTwb and/or RDT, and negative in CATTp (controls).

- Seropositive: positive in CATTwb and/or RDT, and positive in CATTp, but negative in parasitological examination (unconfirmed suspects).

- Confirmed: positive in CATTwb and/or RDT, and positive in CATTp, and positive in parasitological examination (cases).

All confirmed cases (CATTp ≥1/4 with parasitological confirmation) and all unconfirmed seropositive individuals (CATTp ≥1/4 without parasitological confirmation) were proposed for study enrolment. Seronegative controls were randomly selected, and approximately one age-matched seronegative control for every two seropositive cases were enrolled from the same village. Children under 16 years of age and pregnant women were not included in the study. Each participant was informed about the study's objectives in their own language and provided written informed consent. For participants between 16 to 18 years of age, written

**Table 2. Total number of subjects enrolled per transmission focus, date, visit and group.** Out of the 96 subjects presented in the enrollment column, 55 were followed-up once, out of which 12 were followed-up twice.

| Focus | Date | Population screened | Number of subjects at enrolment | | | | | Number of subjects at follow-up 1 | | | | | Number of subjects at follow-up 2 | | | | |
|---|---|---|---|---|---|---|---|---|---|---|---|---|---|---|---|---|---|
| | | | Seronegative | Seropositive | Stage 1 | Stage 2 | Total | Seronegative | Seropositive | Stage 1 | Stage 2 | Total | Seronegative | Seropositive | Stage 1 | Stage 2 | Total |
| Boffa | December 2018 | 2892 | 3 | 12 | 0 | 11 | 26 | 0 | 0 | 0 | 0 | 0 | 0 | 0 | 0 | 0 | 0 |
| | December 2019 | 2924 | 9 | 11 | 1 | 2 | 23 | 1 | 9 | 1 | 8 | 19 | 0 | 0 | 0 | 0 | 0 |
| | December 2020 | 1918 | 0 | 1 | 0 | 1 | 2 | 0 | 9 | 1 | 2 | 12 | 1 | 1 | 0 | 3 | 5 |
| | Sub-total | 7734 | 12 | 24 | 1 | 14 | 51 | 1 | 18 | 2 | 10 | 31 | 1 | 1 | 0 | 3 | 5 |
| Forecariah | April 2019 | 2477 | 15 | 3 | 8 | 11 | 37 | 0 | 0 | 0 | 0 | 0 | 0 | 0 | 0 | 0 | 0 |
| | October 2020 | 3559 | 0 | 0 | 0 | 0 | 0 | 0 | 0 | 0 | 7 | 7 | 0 | 0 | 0 | 0 | 0 |
| | November 2020 | 2896 | 1 | 5 | 1 | 0 | 7 | 0 | 2 | 4 | 4 | 10 | 0 | 0 | 0 | 0 | 0 |
| | April 2021 | 2250 | 1 | 0 | 0 | 0 | 1 | 0 | 4 | 2 | 1 | 7 | 0 | 0 | 3 | 4 | 7 |
| | Sub-total | 11182 | 17 | 8 | 9 | 11 | 45 | 1 | 6 | 6 | 12 | 24 | 0 | 1 | 3 | 4 | 7 |
| Total | | 18916 | 29 | 32 | 10 | 25 | 96 | 1 | 24 | 8 | 22 | 55 | 1 | 1 | 3 | 7 | 12 |

informed consent was also obtained from their parents. The data of individual study participants were randomly anonymized with a 4-digit code at enrolment by the MD in charge of the medical campaign.

## Field procedure and sampling

As in [4], at enrolment, each participant underwent an epidemiological interview and a clinical examination, during which dermatological symptoms were assessed by a trained dermatologist, as well as at each subsequent follow-up after enrolment/treatment. This interview and clinical examination were performed for all confirmed cases, all seropositive individuals, and for all seronegative controls. The following epidemiological data were collected: age (in years), sex, clinical history of HAT infections in the family since 2010, and occupational risk (occurrence of any regular activities such as farming, hunting, fishing, wood-cutting and/or mining, during which an individual might be more exposed to tsetse bites). The following general clinical data were also collected during the interview: fever, swollen cervical lymph nodes, weight loss, asthenia, eating disorders, sexual dysfunctions, repetitive headaches, circadian rhythm disruptions and/or any other behavioral changes during the last three months. Dermatological signs of pruritus (skin itch) and dermatitis (skin inflammation) were also investigated, and a careful examination of the entire body was performed, to detect any symptoms that might be related to skin infections.

Finally, a superficial skin snip biopsy was sampled in sterile conditions from the right back shoulder of all enrolled subjects. In this study, the skin region was the same as in Camara *et al.* [4] for comparison, yet the choice of sampling superficial snip biopsies rather than deep punch biopsies was preferred by the HAT-NCP to limit sampling invasiveness and allow a more rapid healing of the lesion. As skin snip biopsy sampling is routinely used in the context of onchocerciasis diagnosis, this choice was also appreciated by the National Ethical Committee of the Republic of Guinea. Biopsies were performed under local anesthesia and were rapidly dressed. Skin snips were then rapidly fixed in 10% neutral buffered formalin for immuno-histochemistry and RNAlater for molecular analyses.

## Immune trypanolysis test

A plasma sample from all participants was used to perform the immune trypanolysis test. This test detects complement-mediated immune responses activated by either the LiTat 1.3, LiTat 1.5 and LiTat 1.6 variable antigen types (VAT), or variable surface glycoproteins (VSG), specific for *T. b. gambiense*, as previously described [14].

## Immunohistochemical detection of trypanosomes

Skin snip biopsy samples fixed in formalin and preserved at 4˚C were trimmed and processed into paraffin blocks in the lab. Longitudinal sections of ~3μm were prepared and processed using the Leica Bon Rx detection kit (Leica, Germany), according to the manufacturer's recommendation. Sections were immunolabelled with the *T. brucei*-specific anti-ISG65 antibody which targets the Invariant Surface Glycoprotein 65 (rabbit 1/800; gift from M. Carrington, Cambridge, UK) [15]. For immunolabelling, a horseradish peroxidase-coupled secondary antibody was used. A non-infected West African skin specimen (Tissue Solutions Ltd., UK) and a *T. b. gambiense*-infected mouse skin specimen were also included with the samples as technical negative and positive controls, respectively. Immunostaining images were acquired using an automated Axioscan Z1 slide-scanner (Carl Zeiss, Germany) and analysed using the ZEN 3.7 software (Carl Zeiss, Germany). The slides from each biopsy were blindly assessed by at least

two readers. The positivity of a given skin-section slide was defined by the detection of at least five clearly distinguishable trypanosomes in the dermis.

### DNA extraction

Skin snip biopsy samples in RNAlater and preserved at -20°C were treated for DNA extraction in the lab. A non-infected West African skin specimen (Tissue Solutions Ltd., UK) and a *T. b. gambiense*-infected mouse skin specimen were also included with the samples as negative and positive technical controls, respectively. For blood samples, DNA extraction was performed on 1ml blood aliquots. Total DNAs were extracted with DNeasy Blood and Tissue kits (Qiagen, Germany) following the manufacturer's recommendations.

### PCR detection of trypanosome DNA

PCR detection of *T. brucei* parasites was performed using published primers, which hybridize to a 177bp DNA satellite repeat sequence (10,000 copies per cell) to generate a 117bp amplicon as previously described [16]. All PCR results were confirmed using new primers (TBRN3-F 5'-TAAATGGTTCTTATACGAATGA-3' and TBRN3-R 5'-TTGCACACATTAAACACTAAA-GAACA-3') that externally flank the first primer pair to generate a larger fragment of 168bp. Single-round *T. b. gambiense*-specific PCR directed against the single copy *TgsGP* gene was also performed using published primers (308bp amplicon), as previously described [17].

### Data analyses

The complete and anonymized raw data base resulting from the entire study is available in S1 Table. General descriptive analyses of anonymized data were performed using Excel 16.80 (Microsoft, USA). Statistical analyses were performed using Prism V10.1.1 (GraphPad, USA) software. For epidemiological, clinical, and diagnostic parameters, differences between sero-negative controls versus unconfirmed seropositive individuals and confirmed cases were assessed using the following two-sided tests at 5% confidence: Fisher's exact tests for qualitative data and/or Mann-Whitney tests for quantitative data. For the follow-up analyses, differences between results at enrolment versus results at 6 to 12 months after treatment/enrolment were assessed for each group using two-sided Fisher's exact tests at 5% confidence.

## Results

### Epidemiological and clinical results

Results of the initial screening of 18,916 individuals are shown in Table 2. Out of 18,849 sero-negative subjects (CATTwb-negative or CATTp < 1/4), 29 subjects with no exclusion criteria were enrolled in the study (seronegative group). A total of 67 individuals tested positive in CATT (CATTwb-positive and CATTp ≥ 1/4) and with no exclusion criteria accepted to be enrolled in the study, of which 32 tested negative upon parasitological examination (0.17%, seropositive group) and 35 were confirmed as HAT cases (0.19%, confirmed group). In total, 96 subjects were enrolled in the study, including 51 from Boffa (53%) and 45 from Forecariah (47%) (Table 2).

As shown in Table 3, no significant differences between groups were observed for any epidemiological parameters. After the presence of swollen lymph nodes, the occurrence of dermatitis was significantly more frequent in stage 2 cases (16/24, 67%, P = 0.0007) and seropositive individuals (16/32, 50%, P = 0.0152) as compared to seronegative subjects (5/27, 19%). Pruritus was the most frequent dermatological sign in confirmed stage 2 patients (22/25, 88%), as

**Table 3. Epidemiological and clinical characteristics of case subjects.**

| Parameters | | Seronegative (n = 29) | Seropositive (n = 32) | | Stage 1 (n = 10) | | Stage 2 (n = 25) | |
|---|---|---|---|---|---|---|---|---|
| | | n (%) or mean (SD) | n (%) or mean (SD) | p values | n (%) or mean (SD) | p values | n (%) or mean (SD) | p values |
| **Epidemiological** | Age* | 37 (11) | 41 (16) | *0.5969* | 37 (14) | *0.9999* | 32 (13) | *0.3546* |
| | Sex (Male) | 14/29 (48%) | 12/32 (38%) | *0.4449* | 6/10 (60%) | *0.7164* | 9/25 (36%) | *0.4170* |
| | Occupational risk | 19/29 (66%) | 19/32 (59%) | *0.7919* | 8/10 (80%) | *0.6927* | 14/25 (56%) | *0.5789* |
| | HAT case(s) in the family | 11/29 (38%) | 10/32 (31%) | *0.6020* | 7/10 (70%) | *0.2819* | 9/25 (36%) | *>0.9999* |
| **Clinical** | Swollen LN | 6/26 (23%) | 16/31 (52%) | ***0.0329*** | 5/10 (50%) | *0.1087* | 23/25 (92%) | ***<0.0001*** |
| | Pruritus | 12/27 (44%) | 22/32 (69%) | *0.0705* | 7/10 (70%) | *0.1552* | 22/25 (88%) | ***0.0006*** |
| | Dermatitis | 5/27 (19%) | 16/32 (50%) | ***0.0152*** | 5/10 (50%) | *0.0871* | 16/24 (67%) | ***0.0007*** |
| | Weight loss | 9/29 (31%) | 16/32 (50%) | *0.1929* | 7/10 (70%) | *0.0598* | 15/25 (60%) | *0.0540* |
| | Fever | 18/29 (62%) | 26/32 (81%) | *0.1522* | 8/10 (80%) | *0.4451* | 22/25 (88%) | *0.0595* |
| | Circadian rhythm disruptions | 5/29 (17%) | 14/32 (44%) | ***0.0305*** | 6/10 (60%) | ***0.0167*** | 9/25 (36%) | *0.1343* |
| | Eating disorders | 9/29 (31%) | 16/32 (50%) | *0.1929* | 5/10 (50%) | *0.4459* | 10/25 (40%) | *0.5734* |
| | Headache | 18/29 (62%) | 25/32 (78%) | *0.1713* | 7/10 (70%) | *0.7212* | 15/25 (60%) | *>0.9999* |
| | Asthenia | 18/29 (62%) | 25/31 (81%) | *0.1538* | 6/10 (60%) | *>0.9999* | 16/25 (64%) | *>0.9999* |
| | Behaviour changes | 10/29 (34%) | 12/32 (38%) | *>0.9999* | 3/10 (30%) | *>0.9999* | 9/25 (36%) | *>0.9999* |
| | Sexual dysfunctions | 9/28 (32%) | 9/30 (30%) | *>0.9999* | 3/10 (30%) | *>0.9999* | 7/24 (30%) | *>0.9999* |

For each group and each parameter, total values correspond to the numbers of subjects for which a value was available (n/total). p values were obtained by comparing one by one the parameters of each group of seropositive subjects (unconfirmed and all confirmed) to those of seronegative controls using two-sided Fisher's exact tests or * two-sided Mann-Whitney tests at 5% confidence. LN: lymph nodes.

compared to seronegative subjects (12/27, 44%). These observations confirm the importance of dermatological signs in the clinical picture of gHAT.

## Biological results

Plasmas were assessed using the trypanolysis test, which detects complement-mediated immune responses activated by *T. b. gambiense*-specific antigens. In total, 72% (18/25) of stage 2 cases were positive for the three LiTat antigens (Table 4). Only 9% (3/32) of the seropositive individuals were positive for all antigens, while 81% (26/32) remained negative for all three variants, as all seronegative controls (0/27).

The skin snip biopsy samples were processed for immunohistochemistry analyses (IHC) (Fig 2). Skin samples obtained from the seronegative controls (12/12) did not test positive for

**Table 4. Serological, molecular and histological analysis results from blood and skin samples.**

| Parameters | | Seronegative (n = 27) | Seropositive (n = 32) | | Stage 1 (n = 10) | | Stage 2 (n = 25) | |
|---|---|---|---|---|---|---|---|---|
| | | n/total (%) | n/total (%) | p values | n/total (%) | p values | n/total (%) | p values |
| **Trypanolysis** | LiTat 1.3 positive | 0/27 (0%) | 6/32 (19%) | *0.0267* | 7/10 (70%) | *<0.0001* | 23/25 (92%) | *<0.0001* |
| | LiTat 1.5 positive | 0/27 (0%) | 6/32 (19%) | *0.0267* | 8/10 (80%) | *<0.0001* | 23/25 (92%) | *<0.0001* |
| | LiTat 1.6 positive | 0/27 (0%) | 3/32 (9%) | *0.2425* | 2/10 (20%) | *0.0676* | 20/25 (80%) | *<0.0001* |
| | Positive for all VATs | 0/27 (0%) | 3/32 (9%) | *0.2425* | 2/10 (20%) | *0.0676* | 18/25 (72%) | *<0.0001* |
| **TBR-PCR** | Positive on blood | 0/27 (0%) | 3/32 (9%) | *0.2425* | 5/8 (62%) | *0.0002* | 17/25 (68%) | *<0.0001* |
| | Positive on skin | 1/27 (4%) | 13/32 (41%) | *0.0014* | 7/10 (70%) | *<0.0001* | 18/25 (72%) | *<0.0001* |
| **Histology** | ISG65 positive | 0/12 (0%) | 11/28 (39%) | *0.0170* | 1/6 (17%) | *0.3333* | 4/20 (20%) | *0.2709* |

For each group and each parameter, total values correspond to the numbers of subjects for which a value was available (n/total). p values were obtained by comparing one by one the parameters of each group of seropositive subjects (unconfirmed and all confirmed) to those of seronegative controls using two-sided Fisher's exact tests at 5% confidence. VAT: variable antigen type; PCR: polymerase chain reaction; ISG65: invariant surface glycoprotein 65.

trypanosomes (Table 4). By contrast, 39% (11/28, P = 0.017) of the unconfirmed seropositive individuals, and 17% (1/6) and 20% (4/20) of the stage 1 and stage 2 cases, respectively, were found to be positive following staining by a *T. brucei*-specific anti-ISG65 antibody (Table 4).

To confirm the identity of these skin-dwelling parasites, *Trypanozoon*-specific PCR (TBR-PCR) assays were performed on total DNA extracted from blood and skin snip samples. Both blood and skin DNA samples from the 27 seronegative controls were found to be negative by the TBR-PCR assay, excepted one individual positive to TBR-PCR in skin (also with CATTp 1/2, swollen lymph nodes, pruritus, asthenia, and fever). By contrast, 62% (5/8) and 68% (17/25) of the blood samples, and 70% (7/10) and 72% (18/25) of the skin samples from

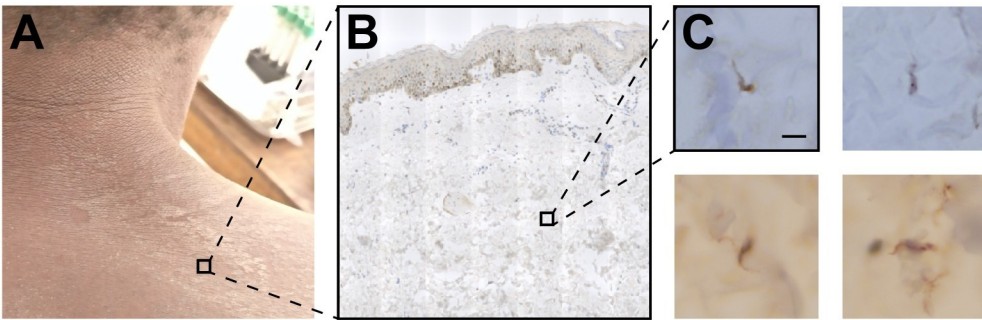

**Fig 2. Dermal trypanosomes evidenced in human skin sections.** Skin snip biopsies were sampled on the right back shoulder of enrolled individuals (A). Formalin-fixed paraffin-embedded skin sections were immunolabelled with the *T. brucei*-specific anti-ISG65 antibody. Immunostaining images were acquired using an automated Axioscan Z1 slide-scanner (B). The darker band on the top of the section in B corresponds to the melanized epidermis, and the white region to the dermis. Note that this section was among the thickest obtained in this study. The positivity of a given skin-section slide was defined by the detection of at least five clearly distinguishable trypanosomes in the dermis (C). The scale bar shows 10μm.

stage 1 and stage 2 cases, respectively, tested positively in the TBR-PCR assay. Interestingly, parasite DNA was mostly detected in the skin of unconfirmed seropositive individuals (13/32, 41%) as compared to blood (3/32, 9%) (Table 4). *T. b. gambiense*-specific TgsGP-PCR assays were performed on the same DNA samples: only 2 blood samples (one stage 1 and one stage 2 cases) and 1 skin sample (from the same stage 2 subject) tested positive.

In total, 50% of the subjects with positive results in IHC were also found positive in TBR-PCR in skin. Moreover, among subjects presenting at least one dermatological sign, 59% (13/22), 71% (5/7), and 86% (19/22) of the seropositive, stage 1 and stage 2 individuals, respectively, were also found positive in the skin by at least one test. In contrast, this was only observed in 8% (1/13) of the seronegative controls. These results suggest that the presence of one dermatological sign is highly correlated with an active HAT infection.

## Follow-up results

Among the 96 subjects included in the study, 55 individuals (34%) were followed-up at 6 to 12 months, and 12 individuals (7%) at 18 to 24 months after enrolment (Table 2). As the follow-up was optional in this study, the decreasing number of subjects for follow-up was due to the unavailability of the subjects or to their refusal to participate in the follow-up procedure. The same panel of analyses were repeated in 55 and 12 subjects at 6–12 months and 18–24 after enrolment, respectively (Table 2). Most of the clinical symptoms observed at enrolment, including dermatological signs, significantly decreased in frequency during the first 6–12 months after treatment of stage-2 cases, and to a lesser extend in stage-1 cases (Table 5). A similar trend was observed for CATT results after treatment in confirmed cases, yet with a CATTp

**Table 5. Clinical, parasitological, serological, molecular and histological follow-up analyses at 6–12 months after enrolment.**

| Parameters | | Seropositive (n = 24) | | | Stage 1 (n = 8) | | | Stage 2 (n = 23) | | |
|---|---|---|---|---|---|---|---|---|---|---|
| | | Enrollment | Follow-up 1 | | Enrollment | Follow-up 1 | | Enrollment | Follow-up 1 | |
| | | n/total (%) | n/total (%) | *p values* | n/total (%) | n/total (%) | *p values* | n/total (%) | n/total (%) | *p values* |
| **Clinics** | Swollen LN | 13/23 (57%) | 6/24 (25%) | *0.0392* | 4/8 (50%) | 1/8 (13%) | *0.2821* | 20/22 (91%) | 2/20 (10%) | *<0.0001* |
| | Pruritus | 15/24 (63%) | 5/23 (22%) | *0.0077* | 5/8 (63%) | 2/8 (25%) | *0.3147* | 19/22 (86%) | 3/22 (14%) | *<0.0001* |
| | Weight loss | 11/24 (46%) | 6/24 (25%) | *0.0940* | 5/8 (63%) | 0/8 (0%) | *0.0256* | 14/22 (64%) | 1/22 (5%) | *<0.0001* |
| | Dermatitis | 11/24 (46%) | 6/23 (26%) | *0.2270* | 4/8 (50%) | 2/8 (25%) | *0.6084* | 13/21 (62%) | 2/22 (9%) | *0.0004* |
| | Fever | 19/24 (79%) | 14/24 (58%) | *0.2124* | 6/8 (75%) | 5/8 (63%) | *>0.9999* | 19/22 (86%) | 7/22 (32%) | *0.0005* |
| | Circadian rhythm disruptions | 10/24 (42%) | 4/24 (17%) | *0.1107* | 5/8 (63%) | 2/8 (25%) | *0.3147* | 8/22 (36%) | 1/23 (4%) | *0.0098* |
| **Diagnosis** | CATTwb | 24/24 (100%) | 15/24 (63%) | *0.0016* | 8/8 (100%) | 5/8 (63%) | *0.2000* | 22/22 (100%) | 11/16 (69%) | *0.0087* |
| | CATTp | 24/24 (100%) | 14/23 (61%) | *0.0006* | 7/8 (88%) | 5/8 (63%) | *0.5692* | 22/22 (100%) | 8/16 (50%) | *0.0003* |
| | mAECTbc | 0/21 (0%) | 0/24 (0%) | *>0.9999* | 7/8 (88%) | 0/8 (0%) | *0.0014* | 16/18 (89%) | 0/23 (0%) | *<0.0001* |
| **Trypanolysis** | LiTat 1.3 positive | 3/24 (13%) | 2/20 (10%) | *>0.9999* | 6/8 (75%) | 2/5 (40%) | *0.2929* | 20/22 (91%) | 14/15 (93%) | *>0.9999* |
| | LiTat 1.5 positive | 3/24 (13%) | 2/20 (10%) | *>0.9999* | 6/8 (75%) | 2/5 (40%) | *0.2929* | 20/22 (91%) | 10/15 (67%) | *0.0953* |
| | LiTat 1.6 positive | 1/24 (4%) | 1/20 (5%) | *>0.9999* | 1/8 (13%) | 2/5 (40%) | *>0.9999* | 17/22 (77%) | 11/15 (73%) | *>0.9999* |
| | Positive for all VATs | 1/24 (4%) | 1/20 (5%) | *>0.9999* | 1/8 (13%) | 1/5 (20%) | *>0.9999* | 15/22 (68%) | 8/15 (53%) | *0.4933* |
| **TBR PCR** | Blood | 2/24 (8%) | 1/24 (4%) | *>0.9999* | 4/6 (67%) | 1/5 (20%) | *0.2424* | 15/22 (68%) | 0/20 (0%) | *<0.0001* |
| | Skin | 8/24 (33%) | 4/23 (17%) | *0.3177* | 6/8 (75%) | 2/8 (25%) | *0.1319* | 16/22 (73%) | 3/22 (14%) | *0.0002* |
| **Histology** | IHC ISG65 | 9/22 (41%) | 4/16 (25%) | *0.4898* | 1/4 (25%) | 0/2 (0%) | *>0.9999* | 3/18 (17%) | 2/10 (20%) | *>0.9999* |

Total values correspond to the numbers of subjects for which a value was available (n/total). For each group of subjects, p values were obtained by comparing one by one the parameters recorded at 6–12 months after treatment/enrolment to those obtained at enrolment, using two- sided Fisher's exact tests at 5% confidence. LN: lymph nodes; CATT: card agglutination test for trypanosomiasis; VAT: variable antigen type; PCR: polymerase chain reaction; ISG65: invariant surface glycoprotein 65; IHC: immuno-histochemistry.

positivity rate maintained at 63% (5/8) and 50% (8/16) in stage-1 and stage-2 patients, respectively. This persisting seropositivy was confirmed by the trypanolysis test results that did not significantly change 6–12 months after treatment (Table 5). Results for parasitological observations (mAECTbc) and PCR on blood and skin became negative within 6–12 months after treatment in most confirmed HAT cases, especially in stage-2 patients (Table 5). In unconfirmed seropositive individuals, positivity rates in PCR on blood (8% to 4%) and skin (33% to 17%) remained relatively stable between enrolment and follow-up at 6–12 months (Table 5). No significant variation was observed in any group by IHC between the enrolment and follow-up at 6–12 months (Table 5). Among stage-1 patients followed-up at 6–12 months after treatment (Table 5): the subject remaining positive in PCR on blood was also still presenting a CATTp positive at 1/4 dilution; one individual positive in PCR on skin was also still presenting a CATTp positive at 1/4 dilution, and the second patient positive in PCR on skin was positive in trypanolysis and with a CATTp positive at 1/4 dilution. Regarding stage-2 patients followed-up at 6–12 months after treatment (Table 5): among the 3 subjects positive in PCR on skin, two were also found positive in trypanolysis and with a CATTp positive at 1/2 dilution, and the third one was positive in trypanolysis, IHC and with a CATTp positive at 1/8 dilution. One of the two stage-2 patients positive in IHC was also found positive in trypanolysis and with a CATTp positive at 1/16 dilution. A total of 12 subjects were followed-up after 18–24 months (S2 Table): all the 3 stage-1 and 7 stage-2 patients tested negative on both blood and skin at the second follow-up, whereas one of the two unconfirmed seropositive individuals tested positive for dermal parasites by IHC and PCR. Interestingly, the subject #1147 (S2 Table) was initially considered seronegative at baseline despite a positivity in CATTwb and the occurrence of characteristic clinical signs (swollen lymph nodes and pruritus). At follow-up 1, this subject was tested positive in CATTp and in IHC, with a WBC count in CSF at 12, but she/he remained negative in parasitology. This highlights the importance of considering clinical signs for diagnosis, as this kind of atypical situation may become more frequent with the decrease of standard cases showing clear parasitological results. The subject #1149 was treated with fexinidazole as a confirmed stage 2 at baseline. The subject tested negative in CATTp, parasitology, trypanolysis, and PCR at follow-up 1, yet she/he was found positive in CATTp (1/32) and mAECT at follow-up 2 and treated with NECT. Unfortunately, the nature of this case (relapse Vs. reinfection) was not investigated further and the skin snip samples were not exploitable.

## Foci comparison

All epidemiological, clinical, parasitological, and biological parameters were compared between subjects from the two foci (Table 6). The occurrence rates of reported circadian rhythm disruptions (64%) and eating disorders (64%) were higher in stage-2 cases from Forecariah as compared to those from Boffa (14% and 21%, respectively), and the occurrence of pruritus more frequent in seropositive unconfirmed subjects from Forecariah (100% Vs. 58%) (Table 6). The most significant differences were the higher proportions of unconfirmed seropositive individuals (88%) and stage-2 confirmed cases (100%) found positive in PCR on skin in Forecariah as compared to subjects from the same goups in Boffa (25% and 50%, respectively) (Table 6).

## Discussion

To extend our previous observations [4], we investigated whether *T. b. gambiense* parasites might be found in the skin of individuals in a large cohort of subjects and in two different active gHAT foci in Guinea (Forecariah and Boffa).

**Table 6. Clinical observations and test results presenting statistically different levels between the two transmission foci.**

| Parameters | | HAT Focus | Seronegative | | Seropositive | | Stage 2 | |
|---|---|---|---|---|---|---|---|---|
| | | | n/total (%) | p values | n/total (%) | p values | n/total (%) | p values |
| **Clinical** | Asthenia | Boffa | **11/12 (92%)** | *0.0080* | | | | |
| | | Forecariah | 7/17 (41%) | | | | | |
| | Behaviour changes | Boffa | **7/12 (58%)** | *0.0460* | | | | |
| | | Forecariah | 3/17 (18%) | | | | | |
| | Circadian rhythm disruptions | Boffa | | | | | 2/14 (14%) | *0.0168* |
| | | Forecariah | | | | | **7/11 (64%)** | |
| | Eating disorders | Boffa | | | | | 3/14 (21%) | *0.0486* |
| | | Forecariah | | | | | **7/11 (64%)** | |
| | Pruritus | Boffa | | | 14/24 (58%) | *0.0353* | | |
| | | Forecariah | | | **8/8 (100%)** | | | |
| **TBR PCR** | Blood | Boffa | | | 0/24 (0%) | *0.0019* | | |
| | | Forecariah | | | **4/8 (50%)** | | | |
| | Skin | Boffa | | | 6/24 (25%) | *0.0032* | 7/14 (50%) | *0.0078* |
| | | Forecariah | | | **7/8 (88%)** | | **11/11 (100%)** | |

Total values correspond to the numbers of subjects for which a value was available (n/total). For each group of subjects, p values were obtained by comparing one by one the parameters recorded in Boffa and Forecariah, using two-sided Fisher's exact tests at 5% confidence. All the other parameters presented in this study were not found to differ significantly between the two transmission foci. PCR: polymerase chain reaction.

Here, we also observed some significant proportions of both confirmed cases and unconfirmed seropositive subjects with extravascular trypanosomes in the skin, demonstrating that this phenomenon is not restricted to Forecariah, where the previous study took place. However, the frequency of dermal trypanosome carriage was apparently lower in Boffa as compared to Forecariah. This could possibly be explained by the different epidemiological status of the two foci, with Boffa being more advanced in disease control (vector control launched in 2012, 9 new cases reported in 2020, only one stage 1 case confirmed in this study), hence with a theoretically lower parasite circulation, as compared to Forecariah (vector control launched in 2018, 21 new cases reported in 2020, nine stage 1 cases confirmed in this study).

These results highlight again dermatological symptoms as an important aspect of gHAT's clinical presentation. As reported in previous studies [4, 18, 19], we observed a higher occurrence of pruritus and dermatitis in seropositive individuals and confirmed cases, relative to seronegative individuals, and this higher frequency of dermatological symptoms was correlated with the detection of trypanosomes in the skin.

Nevertheless, the proportion of skin samples found positive for trypanosomes (41% in seropositive individuals and 71% in confirmed cases) was lower than our previous observations (100% of both groups in [4]). This difference in the sensitivity of immuno-histological and

molecular detection methods between the two studies could likely be explained by the different skin sampling methods. Although the skin region was the same as in Camara *et al*. [4], superficial snip biopsies were preferred to deep punch biopsies to limit surgical invasiveness and allow a theoretically more rapid healing of the lesion. As skin snip biopsy sampling is routinely use in the context of onchocerciasis diagnosis, this choice was also appreciated by the National Ethical Committee of the Republic of Guinea. However, the biological material obtained by snip skin biopsy necessarily contains less dermal tissue than in a punch biopsy, and therefore potentially less dermal trypanosomes that were shown to be enriched in the basal dermis [2].

As only few reports exist and no gold-standard approaches for detection of trypanosomes in the human skin are available [4, 20], we implemented complementary molecular and immuno-histological methods in parallel, yet with their own specific strengths and weaknesses, as discussed previously [4]. Considering that *T. b. brucei* are non-infectious to humans and killed within a couple of hours by human serum, the dermal parasites detected here, at least in confirmed cases, are likely to be *T. b. gambiense* parasites, as confirmed by the positivity of three TgsGP-PCR assays, including one in the skin. Low parasitemia are frequently encountered in HAT patient from Guinea [10], explaining why only few patients tested positive to the low-sensitive TgsGP-PCR performed on blood, although they were found positive either in cervical lymph nodes or by the mAECT performed on buffy coats (sensitivity of 10 trypanosomes/ml). Similarly, the fluctuation of trypanosome densities in the skin could also explain the differences between results in TBR-PCR and TgsGP-PCR on skin samples.

Here again, the detection of skin-dwelling parasites in a large proportion of the skin snip biopsies sampled from seropositive individuals indicates that skin-dwelling parasites might be present over a considerable proportion of the skin surface. According to historic (reviewed in [21]) and more recent [2, 3] studies in experimental animal models, skin-dwelling parasites could theoretically be detected in almost the entire skin surface, yet with a variable distribution and at variable local densities.

Bloodstream parasite numbers in *T. b. gambiense* infections can periodically fluctuate to less than 100 trypanosomes/ml, falling below the detection limit of the most sensitive methods currently in use [22]. It is estimated that 20–30% of gHAT cases are missed in active case detection by standard parasitological techniques and are left untreated [23]. This could explain (i) why one individual with swollen lymph nodes, pruritus, asthenia, and fever in the seronegative group (CATTp1/2) was found positive to TBR-PCR in skin, and (ii) why 3 seropositive individuals with negative confirmatory results in parasitological examination of blood were found positive in TBR-PCR on blood (Table 4). This could also explain the differential negativation dynamics after treatment of confirmed cases according to the sample types and detection methods, with skin parasites being detectable for longer as compared to blood parasites (Table 5). A delay in the trypanocidal effects of the drugs on the extravascular parasites as compared to the blood forms would also be in line with the long-lasting CATT seropositivity observed in patients after treatment.

As recently observed in Cameroon [24], one possible explanation for the persistence of disease foci in certain regions is the presence of animal reservoirs carrying trypanosomes in both blood and skin [25]. Another possibility is that traditionally used diagnostic approaches based on blood examination do not detect some *T. b. gambiense* infections among seropositive cases [25]. Mathematical modelling predicted that, in the absence of any animal reservoirs, these unconfirmed seropositive individuals could contribute to disease transmission by maintaining an overlooked reservoir of skin-dwelling parasites [6], which would slow down progress towards elimination [26]. Hence, these results raise questions about the strategies used to diagnose this disease. The detection of skin-dwelling trypanosomes would (i) allow more carriers

to be treated, (ii) help to estimate the true prevalence of the disease more accurately, and (iii) provide a useful test of cure.

Requiring both costly and fragile equipment and skilled personnel, the detection of trypanosomes in skin biopsies by immunological staining is not suited to active or passive screening strategies implemented in endemic foci. It is nevertheless useful for epidemiological studies and clinical trials. Molecular methods to detect skin-dwelling trypanosomes are likely more adapted and easier to set-up in reference laboratories located in or close to active foci. Considering that the TgSGP-PCR lacks sensitivity and that the TBR-PCR is not specific to *T. b. gambiense*, other molecular methods such as SHERLOCK [27] or multiplex q-PCR [28] are currently under evaluation and may provide more-adapted tools in the near future. The development of less invasive and field-adapted diagnostic methods to detect extravascular dermal trypanosomes, such as the serological detection of skin-related biomarkers or the identification of specific bio-physical profiles by skin scanning also constitute new innovative diagnostic tools that could improve the detection of dermal trypanosomes in endemic populations.

The current WHO recommendation, based on risk-benefit analyses, is to not treat unconfirmed seropositive individuals without knowing if they have an active infection [29]. Importantly, we observed that the routinely administered trypanocide treatments (Pentamidine, NECT and Fexinidazole) efficiently targeted both bloodstream and dermal trypanosomes in all the patients followed-up over 24 months. With the promise of Acoziborole, a new and less toxic drug that only requires a single oral administration, the policy of treating unconfirmed seropositive individuals bearing dermal trypanosomes could possibly be reconsidered [13].

## Supporting information

**S1 Table. Complete raw data base of the present study.**
(XLSX)

**S2 Table. Clinical, parasitological, serological, molecular and histological data from the twelve subjects with 3 visits.** CATTwb / CATTp: card agglutination test for trypanosomiasis on whole blood / plasma; TL: trypanolysis test; mAECT BC / LN aspirate: mini anion-exchange column technique on buffy coat / lymph node aspirate; WBC: white blood cells; CSF: cerebrospinal fluid; PCR: polymerase chain reaction; ISG65: invariant surface glycoprotein 65; IHC: immuno-histochemistry.
(XLSX)

## Acknowledgments

We thank M. Carrington (Cambridge, UK) for providing the anti-ISG65 antibody. We warmly thank the team of the Programme National de Lutte contre la Trypanosomiase Humaine Africaine of Guinea, as well as all our collaborators of the Forecariah and Boffa District Health Departments. We thank Dominique N'Diaye (IPGui, Guinea) for his critical reading of the manuscript.

## Author Contributions

**Conceptualization:** Annette MacLeod, Jean-Mathieu Bart, Mamadou Camara, Bruno Bucheton, Brice Rotureau.

**Data curation:** Oumou Camara, Aline Crouzols, Aïssata Camara, Brice Rotureau.

**Funding acquisition:** Vincent Jamonneau, Annette MacLeod, Jean-Mathieu Bart, Mamadou Camara, Bruno Bucheton, Brice Rotureau.

**Investigation:** Alseny M'mah Soumah, Mariame Camara, Justin Windingoudi Kaboré, Ibrahim Sadissou, Hamidou Ilboudo, Christelle Travaillé, Oumou Camara, Magali Tichit, Jacques Kaboré, Salimatou Boiro, Aline Crouzols, Jean Marc Tsagmo Ngoune, Aïssata Camara, Jean-Mathieu Bart, Bruno Bucheton, Brice Rotureau.

**Methodology:** Vincent Jamonneau, Annette MacLeod, Jean-Mathieu Bart, Mamadou Camara, Bruno Bucheton, Brice Rotureau.

**Project administration:** Jean-Mathieu Bart, Mamadou Camara, Bruno Bucheton, Brice Rotureau.

**Supervision:** Jacques Kaboré, David Hardy, Jean-Mathieu Bart, Mamadou Camara, Bruno Bucheton, Brice Rotureau.

**Writing – original draft:** Brice Rotureau.

**Writing – review & editing:** Hamidou Ilboudo, Christelle Travaillé, Jean Marc Tsagmo Ngoune, David Hardy, Aïssata Camara, Vincent Jamonneau, Jean-Mathieu Bart, Bruno Bucheton, Brice Rotureau.

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
