## [Decision Letter · Decision Letter 0]

17 Jul 2024

Dear Dr. Rotureau, dear Brice

Thank you very much for submitting your manuscript "Prevalence of dermal trypanosomes in suspected and confirmed cases of gambiense human African trypanosomiasis in Guinea." for consideration at PLOS Neglected Tropical Diseases. As with all papers reviewed by the journal, your manuscript was reviewed by members of the editorial board and by several independent reviewers. The reviewers appreciated the attention to an important topic. Based on the reviews, we are likely to accept this manuscript for publication, providing that you modify the manuscript according to the review recommendations. 

Reviewer 1 is supportive of the manuscript, highlighting its clear presentation, relevance to public health, and implications for HAT elimination.

Reviewer 2 also supports the manuscript, emphasizing its importance in identifying skin parasites in HAT cases and suggesting minor revisions for clarity.

Reviewer 3 expresses some concerns about the similarity of this study to the authors' previous work published in Clinical Infectious Diseases, noting that the core message and data might be very similar.

Editorial Evaluation:

The manuscript unveils important findings concerning the detection of trypanosomes in the skin of HAT cases, offering insights that could influence disease management and strategies for elimination. Reviewer 3's concerns regarding similarities to a previous publication are somewhat valid, indicating that while the findings are significant, they may not represent a substantial advancement beyond existing literature. However, I am confident that the authors can effectively address this concern in the revised manuscript.

Recommendations for Revision:

Address Novelty: Clearly articulate how this study advances beyond the authors' previous work in Clinical Infectious Diseases. Highlight any new findings, methodologies, or insights that distinguish this study.

Clarity and Interpretation: Clarify definitions (e.g., seropositive non-confirmed individuals), improve labeling in figures, and provide clearer interpretations of results, as suggested by Reviewer 2.

Comparative Analysis: Address why certain clinical parameters are not statistically significant and compare with previous studies, as requested by Reviewer 2.

Data Availability: Consider making individual patient data available for transparency, as suggested by Reviewer 2.

Conclusion:

While the manuscript presents valuable findings on the presence of trypanosomes in the skin and its implications for HAT control, it is crucial to address concerns regarding novelty and overlap with previous publications. The editors encourage the authors to revise the manuscript to clearly differentiate it from their previous work and to enhance clarity and interpretation of the findings. 

Sincerely,

Markus Engstler

Guest Editor

Hira Nakhasi

Section Editor

Reviewer's Responses to Questions

**Key Review Criteria Required for Acceptance?**

**Methods**

-Are the objectives of the study clearly articulated with a clear testable hypothesis stated?

-Is the study design appropriate to address the stated objectives?

-Is the population clearly described and appropriate for the hypothesis being tested?

-Is the sample size sufficient to ensure adequate power to address the hypothesis being tested?

-Were correct statistical analysis used to support conclusions?

-Are there concerns about ethical or regulatory requirements being met?

Reviewer #1: The methods are clearly reported and are appropriate for this study.

Reviewer #2: I am not qualified to evaluate the statistical aspects of this study.

Reviewer #3: The methods employed in this study are virtually identical to those employed by the same authors in their previous study (reference 4, Clinical Infectious Diseases 2021;73(1):12–20 ).

The only difference in this study was the use of a different skin sampling procedure, i.e. a skin snip versus a blood-free skin punch biopsy. This study also included another active gHAT focus, Boffa, located about 100 km north west of the previously investigated focus of the Forecariah district of the republic of Gambia. As in this previous study the methods and their description are sound.

**Results**

-Does the analysis presented match the analysis plan?

-Are the results clearly and completely presented?

-Are the figures (Tables, Images) of sufficient quality for clarity?

Reviewer #1: The results were clearly presented.

Reviewer #2: Please read general comment below.

Reviewer #3: The study essentially demonstrates.

1. The presence of live T. gambiense trypanosomes in the skin of confirmed and unconfirmed seropositive individuals is not restricted to the previously investigated focus of Forecariah but is also found in cases at another focus of infection: Boffa.

2. The most significance difference between the two foci was the higher % of individuals found to be positive by TBR PCR on skin in Forecariah compared to Boffa. The lower dermal carriage in the latter group was attributed more advance disease control measures in this region.

3. The % of skin samples positive for trypanosomes (Fig. 4) in both groups of individuals (unconfirmed seropositive and confirmed cases) in compared to the previous study (ref 4). This was attributed to the differences in skin biopsy methods, with the skin snip method being less sensitive source of material for immunohistological methods than the punch biopsy because by its nature it contains less dermal material.

**Conclusions**

-Are the conclusions supported by the data presented?

-Are the limitations of analysis clearly described?

-Do the authors discuss how these data can be helpful to advance our understanding of the topic under study?

-Is public health relevance addressed?

Reviewer #1: The public health relevance is addressed and their results will advance our understanding of this topic.

Reviewer #2: Please read general comment below.

Reviewer #3: The main result is that immunohistological/PCR methods revealed a significant positive testing for infection when applied to skin samples from seropositive but unconfirmed individuals that was not detected when applied to the blood samples. Interestingly, this positivity in skin biopsies correlated well the clinical manifestations of infection (Table 5). So seropositive individuals who did not show infection by typical parasitology tests on blood, were infected and exhibited clinical signs of infection but the problem of a clinical parameter based diagnosis is the comparable high level of positivity in the seronegative population when compared to the PCR/IH on blood or skin of the group (essentially zero)

The authors make the point that there is a need to develop less invasive field adapted methods to detect extravasulature trypanosomes given the fact that these compartments are relevant for infection and transmission. On this point the authors may not be aware of a recent publication, (Larcombe SD et al. , 2023, PNAS v120, e2306848120) which shows that proliferating parasites are apparently scarce in the blood after infections are established and that this population might overwhelmingly adapted for transmission. It might be worth considering the relevance of issue of slender/stumpy forms in terms of dermal v bloodstream populations. It is hard to say but the images in Fig 2 look like slender forms. Why not try PCR on a stumpy marker such as PAD1, this might add another element to the results to distinguish them from their previous publications.

**Editorial and Data Presentation Modifications?**

Reviewer #1: I have a few minor points that I spotted when reviewing the manuscript.

Line 53-54 – It was not clear how the lower sensitivity of skin snips was critical for detecting trypanosomes in the basal dermis – can you rephrase?

Line 140 – should this read skin snip biopsy not skin snip punch biopsy?

Line 241 – delete ‘of prospections’

Line 241 – should be ‘participate in’

Line 245 – should be ‘to a lesser extent’

Line 260 – should be ‘3 subjects positive’

I couldn’t find a figure legend associated with the image of the hospital and the additional images of the trypanosomes found by histology.

Reviewer #2: This paper describes a significant prospective observational cohort study aimed at assessing the presence of parasites in the dermis of humans living in areas where HAT transmission remains detectable. This is an important question because, to achieve HAT elimination as established by the WHO, it is vital to identify and monitor all parasite reservoirs. A previous study by these authors demonstrated that T. b. gambiense could be detected in the skin of patients (10.1093/cid/ciaa897). Here, the authors provide a larger-scale prospective observational study (initial screening of 18,916 individuals) in two regions of Guinea.

The authors collected clinical, epidemiological, and parasitology data, allowing the classification of individuals as seronegative, seropositive, stage 1, or stage 2. Multiple tests were used to detect parasites, including PCR, immunohistochemistry, CAT assay, and trypanolysis. The authors observed that among confirmed cases, 71% showed parasites in the skin, and among seropositive non-confirmed cases, this number was still quite high (41%). These results suggest that the presence of parasites in the dermis is common, and even when parasites are not detected in the blood, they may be present in the skin. Similar observations have been recently described for extravascular trypanosomes in animals (DOI: 10.1186/s13071-024-06277-7). Importantly, 6-12 months after the initial screening and treatment of all confirmed cases, 25% of seropositive non-confirmed cases still showed parasites in the skin. These findings suggest that, diagnostic methods should be improved to detect skin parasites, and perhaps treatment of seropositive individuals should be considered.

I have no major concerns regarding the research described here. Below are minor points to improve clarity:

- Could the authors label the histology images to indicate the epidermis and the dermis?

- For a non-specialist in the diagnosis and treatment of HAT cases, it is not clear what a seropositive individual is (someone infected in the last X months; is X known?). It should also be clearer in the Results section that a seropositive individual without detectable parasites in the blood and CSF is considered “seropositive non-confirmed” and is not treated. This is important for understanding the implications of the various numbers described throughout the text.

- I recommend that in the Results section, description of data is followed by suggestions of its meaning. For example, the paragraph 232-236 could end with a sentence stating, “These results suggest that the presence of one dermatological sign is highly correlated with an active HAT infection or seropositive non-confirmed.”

- In the Discussion, could the authors comment on why most clinical parameters are not statistically significant and how this compares with previous studies (doi:10.1371/journal.pntd.0002003)?

- Will data form individual patients be available for download? Table S1-S3 has only one tab.

Thank you for this important work and for considering the points above.

Reviewer #3: Specific comments on text

Table 2, a legend would be helpful. The follow up 1 & 2 details are described in the results but should be included here too. In the results the authors mention a decreasing number of subjects for follow up and give reasons for it. However, the figures in the Table are potentially confusing. For example the total number of seropositive unconfirmed (32) and confirmed stage 1 (10) and stage 2 925) add up 67, which is the same number of as the combined totals for follow up 1 (55) and 2 (12). This might be confusing. Maybe be clear that the 55 in follow up 1 is out of the total of 67 that were enrolled as either seropositive and confirmed, and that this dropped to only 12 ex 67 in follow up 2.

Table 3. It should be acknowledged that the main clinical parameters, swollen LN, Pruritis, Dermatitis) give almost as good a rate of indication of infection in seropositive unconfirmed individuals as the TBR-PCR and IH on skin samples (table 4) but of course this is qualified by the high rate of similar positivity in the seronegative group. (maybe this is in the text but if so it was no so exlicid).

Table 5 and P11.

The following section was confusing.

Regarding stage-2 patients followed-up at 6-12 months after treatment (Table 5): among the 3 subjects positives in PCR on skin, two were also found positive in trypanolysis and with a CATTp positive at 1/2 dilution, and the third one was positive in trypanolysis, IHC and with a CATTp positive at 1/8 dilution. The second stage-2 patient positive in IHC was also found positive in trypanolysis and with a CATTp positive at 1/16 dilution.

Should this read The “third” stage-2 patient?

More generally are the results for these specific individuals, and similarly for those in stage 1, which are described in the text also contained in the figures for the trypanolysis assay for example for stage 8 ex 15 for all VATs. This suggests that individuals who are not positive by TBR PCR are actually positive for circulating antibodies against the VATs. It is not clear either why the number of individuals screen varies in these cases. For example in the trypanolysis assays for Stage 2 at follow up 1 there were 15 screen but 22 individuals of this group were screen ed in the TBR PCR skin assay.

It is also odd that the numbers given in the enrolment in Table 5, e.g for stage 1 it is 8 when looking at clinical parameter, differs for that given in table 3, where a number of 10 is given for this group. The same is true for other groups of the seropostives ..for example 24 in T5 v 32 in T3. Why is there a difference between these figures, its not clear from the text.

Minor points, it would be helpful to explain briefly what ISG65 is and why it was included.

VAT, maybe just a note to say same as VSG?

**Summary and General Comments**

Reviewer #1: In this manuscript the authors report their findings from an observational cohort study from a region of Guinea that is currently a HAT focus. The authors showed that as previously seen trypanosomes were readily detected in the skin. Moreover, trypanosomes were seen in the skin of both confirmed and unconfirmed HAT cases. This is an important study as it will have implications for the strategy for HAT elimination, which they discussed. Overall, the data is clearly presented and the report well-written.

Reviewer #2: (No Response)

Reviewer #3: It seems to this reviewer that this study is more or less repeats the work described in a previous study same authors ( Ref 4). The results are also similar and support the view that the skin is a significant reservoir of extravascular parasites and that trypanosomes are present in this compartment in undiagnosed/unconfirmed individuals. They also present bring evidence from foci of infections in the field that support, align with ideas of the initiating study that highlighted which identified the skin as a significant reservoir of trypanosomes that goes undetected by typical bloodstream parasite assessments (Ref 2). In turn these skin resident trypanosomes, which are not detectable by the typical bloodstream based diagnostic approaches could contribute to disease transmission.

Overall, the results and the data support the arguments and views of the authors in their discussion. It is also nice to see publications reflecting HAT in the field, so there is a direct relevance to the study for transmission and disease elimination strategies. Having said that, there is no doubt that the results are very similar in content, outcome, and impact to that of a previously published study by the same authors (Ref 4).

PLOS authors have the option to publish the peer review history of their article (what does this mean?). If published, this will include your full peer review and any attached files.

Reviewer #1: No

Reviewer #2: Yes: Luisa M Figueiredo

Reviewer #3: No

Figure Files:

Data Requirements:

Reproducibility:

References

---

## [Editor Report · Decision Letter 1]

5 Aug 2024

Dear Dr Rotureau, dear Brice

We are pleased to inform you that your manuscript 'Prevalence of dermal trypanosomes in suspected and confirmed cases of gambiense human African trypanosomiasis in Guinea.' has been provisionally accepted for publication in PLOS Neglected Tropical Diseases.

Best regards,

Markus Engstler

Guest Editor

Hira Nakhasi

Section Editor

The authors have found an open response to the main criticism raised by one of the reviewers. They make it unequivocally clear that the technical advances in this manuscript, compared to the published article, are indeed present but not decisive. There are also no groundbreaking new insights. Therefore, the paper would normally be rejected. However, the reviewer who raised this point did not take the easy way out but argued very cautiously and openly. The authors explain the limitations of their possibilities very convincingly. It is incredibly difficult to obtain patient samples of a tropical disease that has become very rare nowadays. It might be even harder to analyse these samples with contemporary techniques. Considering that this has been done over a period of four years, the standards for the degree of innovation demanded by an ever-increasing pressure for innovation are considerably put into perspective. In view of this, the progress that the authors present is remarkable. The present revised version meticulously addresses all other points raised by the reviewers, provides new supplementary data, and clears up misunderstandings.

The manuscript is ready to be published as it is in PLoS NTDs.

---

## [Editor Report · Acceptance letter]

12 Aug 2024

Dear Dr Rotureau,

We are delighted to inform you that your manuscript, "Prevalence of dermal trypanosomes in suspected and confirmed cases of gambiense human African trypanosomiasis in Guinea.," has been formally accepted for publication in PLOS Neglected Tropical Diseases.

Best regards,

Shaden Kamhawi

co-Editor-in-Chief

Paul Brindley

co-Editor-in-Chief
